# Lightweight Structural Materials in Open Access: Latest Trends

**DOI:** 10.3390/ma14216577

**Published:** 2021-11-02

**Authors:** David Blanco, Eva María Rubio, Raquel María Lorente-Pedreille, María Ana Sáenz-Nuño

**Affiliations:** 1Department of Manufacturing Engineering, Industrial Engineering School, Universidad Nacional de Educación a Distancia (UNED), St/Juan del Rosal 12, E28040 Madrid, Spain; erubio@ind.uned.es; 2Mechanical Engineering Department, ICAI, Comillas Pontifical University, Alberto Aguilera 25, E28015 Madrid, Spain; rmlorente@comillas.edu (R.M.L.-P.); msaenz@iit.comillas.edu (M.A.S.-N.)

**Keywords:** lightweight, multi-material, polymer, aerospace, automotive, sustainable

## Abstract

The aeronautical and automotive industries have, as an essential objective, the energy efficiency optimization of aircraft and cars, while maintaining stringent functional requirements. One working line focuses on the use of lightweight structural materials to replace conventional materials. For this reason, it is considered enlightening to carry out an analysis of the literature published over the last 20 years through Open Access literature. For this purpose, a systematic methodology is applied to minimize the possible risks of bias in literature selection and analysis. Web of Science is used as a search engine. The final selection comprises the 30 articles with the highest average numbers of citations per year published from 2015 to 2020 and the 7 articles published from the period of 2000–2014. Overall, the selection is composed of 37 Open Access articles with 2482 total citations and an average of 67.1 citations per article/year published, and includes Q1 (62%) and Q2 (8%) articles and proceeding papers (30%). The study seeks to inform about the current trends in materials and processes in lightweight structural materials for aeronautical and automotive applications with a sustainable perspective. All the information collected is summarized in tables to facilitate searches and interpretation by interested researchers.

## 1. Introduction

Sustainability and energy efficiency are currently key objectives in the development of technological and industrial sectors. These objectives are demanded by society and regulated by the different governments and administrations through increasingly restrictive legislation, aiming to reverse, as far as possible, the environmental damage caused so far and to establish the basis for truly sustainable and efficient development.

According to the Intergovernmental Panel on Climate Change (IPCC) reports, the human influence on global warming is undeniable. Carbon dioxide concentrations have increased by 40% since pre-industrial levels because of fossil fuels and land-use changes. The oceans have absorbed about 30% of the carbon dioxide emitted, leading to their acidification. To stop climate change, greenhouse gas emissions need to be significantly and sustainably reduced [1].

The Advisory Council for Aeronautics Research in Europe (ACARE) was founded in Paris in June 2001. This organization includes members from the EU Member States, the European Commission and stakeholders, i.e., manufacturing industry, airlines, airports, service providers, regulators, research centers, and academia. The 2050 targets include a 75% reduction in CO_2_ emissions per passenger kilometer, a 90% reduction in NOx emissions, and a 65% reduction in aircraft noise in flight, using 2000 capabilities as a reference for the reduction assessment. In addition, aircraft movements must be free of taxing emissions, and air vehicles must be designed and manufactured to be recyclable. Within this framework, and to meet current and future requirements, new innovative materials, design techniques, and manufacturing processes are being developed to increase efficiency and reduce consumption [2].

The worldwide aircraft industry generates 2% of all human-induced CO_2_ emissions. Since the 1960s, CO_2_ emissions per seat kilometer have been reduced by 80%, and since 1990, more than 10 billion tons of CO_2_ have been prevented through the application of new technologies and efficiency improvements. In 2008, Airbus committed to the CO_2_ emission targets of the Air Transportation Action Group (ATAG). These targets set a CO_2_ emissions limit for 2050 of half of the 2005 emissions [3]. In 2018, all the airlines in the world operating on international routes began officially monitoring and reporting their emissions as part of the historic Carbon Offsetting and Reduction Scheme for International Aviation (CORSIA), launched by the United Nations in 2016 [4].

On the other hand, it is also important to mention the current crisis in the aviation sector due to COVID-19. The pandemic caused a historically unprecedented collapse in global air passenger traffic. In April 2020, the industry experienced a 94% drop, and by May 2020, 100% of the world’s destinations were under travel restrictions [5].

At the beginning of 2021, three potential scenarios for air traffic recovery were considered, the first assuming a hypothetical distribution of the vaccine to a large percentage of travelers by summer 2021 with an expected recovery of 2019 levels by 2024, the second with the distribution of the vaccine in the same conditions in summer 2022 with an expected recovery of 2019 levels by 2026, and the third with a non-effective vaccine and expected recovery by 2029 [6].

Regarding the automotive industry, passenger cars, vans, motorbikes, and similar vehicles on two and three wheels represent about one-third of world oil demand and produce about half of all transport-related greenhouse gas emissions. Although substantial progress has been made in controlling pollutant emissions from light-duty vehicles, they contribute significantly to local air pollution with carbon monoxide, hydrocarbons, nitrogen oxides, particulate matter, and other toxins [7].

Regarding the shipbuilding industry, the needs and objectives are similar; according to the Energy Efficiency Design Index (EEDI), the regulation requires ships to be 30% more efficient by 2025 [8].

This reduction in CO_2_ and pollutant emissions by transport-related industries can be addressed through different lines of work, such as developing more efficient combustion engines or engines based on new technologies [9] and reducing the mass of cars and aircraft, either by replacing conventional materials with lightweight materials such as advanced polymers, light alloys, and multi-materials [10], or by designing structures with optimized mass-to-mechanical strength ratios [11].

It is estimated that a mass reduction of 100 kg in a car reduces CO_2_ emissions by 5–12 g. However, despite these efforts, in the last two decades, the average mass of light-duty vehicles has increased by 20% in Europe and by 25% in the United States due to the integration of new safety and comfort systems. A 10% reduction in vehicle mass brings reductions in fuel consumption between 5% and 7%. The main lightweight candidates for vehicle bodies are high-strength steels [11,12], aluminum [10,13], magnesium [14,15], and polymers [16,17].

In addition, aeronautical companies are making major efforts to reduce emissions. Boeing’s priorities include the reduction of CO_2_ emissions and improving fuel efficiency as the two most important priorities of Boeing’s environmental strategy. The 737 MAX consumes 20% less fuel than the aircraft that it replaces, and it is considered as efficient as a hybrid electric car in terms of equivalent liters of fuel burned per 100 km and passenger [4]. Airbus, meanwhile, has designed and manufactured the A350 series using 53% advanced lightweight fiber-reinforced polymer (FRP) composites, and it is 25% more fuel-efficient than its predecessors. The A220 series airframe includes Al-Li parts in the fuselage and titanium and FRP components in the wings, and is considered the most fuel-efficient family of aircraft of its class [3].

Another critical aspect of sustainability is the recycling of materials. Over the next 20 years, more than 12,000 aircraft are expected to complete their life cycle, so the use of materials that enable recycling or reuse at the end of an aircraft’s life is critical. Since 2007, Airbus has recycled 117 jets, with 92% reuse of remaining parts and 100 engines fully recycled [18].

Aluminum alloys have an excellent density-to-mechanical properties ratio. Particle-reinforced aluminum matrix MMCs (metal matrix composites) have a good mass-to-strength ratio, good ductility, high strength and elastic modulus, low coefficient of expansion, excellent wear and corrosion resistance, high creep temperature, and good fatigue behavior [13,19], and they are used in the automotive and aerospace industries in applications such as robots, high-speed machinery, high-speed rotating shafts, automotive engines, and brake parts [10].

Carbon fiber-reinforced polymers (CFRPs) are widely applied in the aerospace, shipbuilding, motorsport, and automotive sectors, and their main advantages are that they are lightweight and have high mechanical and corrosion resistance [17]. These materials are manufactured through processes in which a close to net shape is obtained, but often require final machining to achieve the dimensions and geometry specified in the design. Carbon fibers can reach 3000 °C before they begin to degrade [20] and are very abrasive, making the machining process difficult.

Meanwhile, the increasing use of reinforced carbon fibers pushes aircraft manufacturers to switch from aluminum to titanium parts due to the incompatibility between aluminum and carbon [21]. Titanium has exciting properties such as light weight, high wear and corrosion resistance, and the capability to maintain high strength at high temperatures, which is why it is used in industries such as aerospace and biomedical. However, titanium alloy is considered difficult to machine due to its low thermal conductivity, which causes excessive tool wear during machining, increasing the cost of machining [22,23,24].

The development of a new alloy is a long-term process, e.g., gamma alloys required 40 years from the first exploratory research to the first commercial flight using parts engines from this alloy [25]. There is recent literature on alloys applied in various research, such as gamma alloys, Ni–Ti alloys with memory and superelasticity [26], Al–Li alloys [27] with improved lightness and strength (and currently used by Airbus in the fuselage of the A220 model [3]), and the nickel-based alloys present in 50% of aero-engine alloys [28].

On the other hand, high-performance components can no longer be designed using a single material or material type. Today, to optimize the specific properties of each material, it is common to use multiple materials in the manufacture of a component, especially in lightweight material design. These combinations of materials are called hybrid compounds or multi-materials, and they require the optimization of manufacturing processes. For example, in the machining of a multi-material frequently used in the aeronautics industry, the CFRP/Ti composite, the simultaneous machining of two dissimilar materials requires a choice of compromised manufacturing conditions, leading to worse results than machining them individually [29].

The buy-to-fly (BTF) ratio is the ratio between the initial mass of material at the beginning of the manufacturing process and the mass at the end of the final product. In the aerospace industry, its values vary in the range of 10–20, which means that a lot of material is wasted. For this reason, one of the most studied topics is additive manufacturing, as processes such as wire arc additive manufacturing (WAAM) reduce the BTF ratio to up to 1.2 [21,30,31]. The most analyzed additive manufacturing process in the selected literature is selective laser melting (SLM) [26,32,33,34,35].

On the other hand, the aim of the study is not only to conduct a narrative analysis of the state of the art of scientific literature on structural lightweight materials for sustainable purposes, but also to analyse and understand trends. In studies carried out on existing studies, not all publications, studies, or articles offer the same guarantees of veracity, methodological quality, and interest. For this reason, this study applies a methodology adapted from the Preferred Reporting Items for Systematic Reviews and Meta-Analyses (PRISMA) statement [36] to the field of engineering, to limit possible biases in the selection of the literature used in the trend analysis and to have the most relevant and representative articles available, with proven quality and clear and homogeneous inclusion and exclusion criteria. Among the selection criteria used are: the selection of literature only in Open Access as a criterion that aims to democratize knowledge [37,38,39,40]. This is beginning to be used even for the hiring of university professors, to the detriment of previous parameters such as the impact factor of the publications [41,42], and which has a positive effect on the number of citations associated with the studies. English as the language of publication also has an effect on the number of citations [43], and more than 90% of scientific articles are published in English [44]. The average number of citations per publication year has been used as a criterion for filtering the studies that generate most interest, as it is a widely accepted bibliometric indicator [45,46,47]. Web of Science (WoS) was used as the search engine because it is considered the most important repository of scientific papers worldwide [48], together with Scopus, and also because 94% of their scientific citations overlap between the two repositories [47].

On the other hand, it is necessary to point out that the research aims at sustainable mechanical forming processes involving lightweight materials and multi-material materials, and the field is already vast. For this reason, although adhesive bonding is a topic of great relevance and interest in structural lightweight multi-materials, it has not been included in the search equations in order to focus the study on the main objective of the research. However, the literature associated with the adhesives is extensive, and researchers interested in this type of technology can find extensive reviews of the state of the art [49,50,51], research on process optimisation [52,53], on the repair and detection of damage in adhesively bonded composites [54], or on the debonding of adhesively bonded multi-materials [55], among others.

The methodology followed is objective, rigorous, and meticulous, aiming to reach unbiased, truthful, and representative conclusions about the area of study and period analyzed [36,56]. Initially, quality, inclusion, and exclusion criteria are predefined, and Boolean equations are established to search. The Web of Science (WoS) database is defined as the source of information for the search. Inclusion criteria included only articles published in Open Access, in English, in journals with an impact factor of Q1–Q2, or proceedings of well-known conferences. Subsequently, all pre-selected papers are double-checked to ensure that they meet the pre-defined criteria before being included in the final selection. The documents with the highest average number of citations per year published within the defined period are selected.

## 2. Methodology: Trend Analysis in Structural Lightweight Materials

The field of lightweight structural materials is a very vast domain. It includes all plastic materials and light alloys, either individually or as part of multi-material composites. Figure 1a shows a graph of the studies found in the Scopus database, using a general search of papers that include the topic “lightweight materials” and apply to the aeronautical and automotive sectors. As the graph shows, the number of studies increased by a factor of more than five over the last 20 years [57].

The flow chart in Figure 1b graphically depicts the methodology applied for selecting and analyzing the literature. The goal is to obtain a selection of relevant, comparable articles chosen based on pre-established criteria and, thus, to minimize the risk of bias by standardizing all the decisions involved in the process.

### 2.1. Methodology for Article Selection and Analysis

This research reviews and critically analyses the scientific literature published during the period 2000–2020, and mainly focused on the most recent period of 2015–2020, on lightweight structural materials and multi-materials of potential application to the aeronautical and automotive fields, involving machining processes and/or innovative manufacturing processes. A methodology for selecting and analyzing the research data is previously established to minimize any possible bias. Initially, quality, inclusion, and exclusion criteria are pre-defined. Subsequently, a search is carried out using Boolean equations based on the previously defined criteria and an initial pre-selection and a final selection are made.

The WoS database is defined as the search engine. This database, provided by Clarivate Analytics, allows tracking of more than 1.7 billion references cited in more than 159 million records [58]. WoS is the world’s most popular search engine for the analysis of scientific citations and other bibliometric indicators, and supports many scientific research studies in a variety of fields over the last 20 years [45,47,48]. As a comparison criterion, the average number of citations per publication year is used, as it is a criterion frequently used as a bibliometric indicator [45,46,47]. The language of the studies selected was restricted to English-language publications. English is currently the most important language of exchange for scientific studies, as more than 90% of studies are published in English. In addition, the language of publication has an influence on the number of citations, with a higher number of citations for English-language papers [43,47]. Regarding the type of publication, Open Access or non-Open Access, the selection was restricted to papers published in Open Access. Open Access is a trend that aims to democratize knowledge and make it useful and accessible to everyone [39,40]. For example, the University of Utrecht has ceased to use the impact factor and is beginning to use commitment to Open Access as a criterion for employing or evaluating its teaching staff [41]. Open Access has a positive effect on the number of citations.

It is essential to highlight that this is an Open Access systematic review that aims to determine the latest trends by means of a systematic assessment of relevant peer-reviewed papers with demonstrated approval by the research community, published in peer-reviewed journals and renowned congresses, and it is not a ranking of the most relevant papers.

The Boolean Equation (1) shows the search strategy followed. The key target of this search strategy is to identify the scientific literature published on lightweight structural or multi-material materials with applications in the aeronautical or automotive sectors and involvement in machining processes. Field tags allow searching data fields within a record. In this research, TS was chosen to conduct a data search on title, abstract, and author keywords.
TS * = ((aeronaut * OR aircraft OR aerosp * OR air transport OR aviation OR automobile * OR vehicle * OR automotive *) AND (drilling OR turning OR cutting OR machinability OR machining) AND (hybrid compo * OR hybrid material * OR hybrid structure * OR multi-material * OR multimaterial * OR lightweight * OR light-weight * OR light allo * OR light material * OR structural material *) NOT (nano *) AND (magnesium OR aluminum OR titanium OR steel OR FRP OR GFRP OR CFRP OR Metal OR Polymer OR Ceramic OR Polymer OR textile)).
TS * searches for topic terms in the fields title, abstract, and author keywords within a record.(1)

Besides, the search in the WoS database also requires that the studies are published in Open Access and English. These constraints and the search equation make it possible to obtain an initial pre-selection. Once the pre-selection has been acquired, it is necessary to check the compliance with the predefined inclusion/exclusion criteria of each study individually, i.e., articles published in Open Access, in English, in journals with an impact factor of Q1, Q2, or proceedings of recognized conferences, and that the topics defined by the search criteria are actually included.

In the case of structural lightweight materials, the published literature is very extensive, so a selection is made of the 30 articles with the highest average number of citations per year of publication during the period 2015–2020, and of the seven articles with the highest average number of citations from the period 2000–2014. Finally, an overall selection has been obtained for the period 2000–2020 of 37 articles, 23 publications in Q1 (62%), 3 publications in Q2 (8%) and 11 publications in proceedings of well-known conferences (30%), and an average of 67.1 citations/(article x year published). Data for the periods 2000–2014 and 2015–2020 about the average number of citations per article and the scientific journals where the articles have been published are shown separately in Table 1 and Table 2.

Although papers published in 2020 were included in the definition of the search, no study had reached the annual average number of citations required to appear in the final selection at the time of the last WoS search on 3 July 2021. Total citations and years of publication are based on data provided by WoS on 3 July 2021. Averaged citations per year of publication are calculated considering the difference in years for each article between the year of publication and the year 2020, both included.

Once the pre-selected articles have been reviewed and the final selection made, the relevant information is extracted to a database created for this purpose. That includes title, authors, scientific journal with its classification by percentile or congress, year of publication, digital object identifier (DOI), and the number of total and average citations. In addition, univocal questions are defined and proposed to each article to obtain an overview of interests and trends in the topics and materials investigated.

A summary of the pre-established criteria for the search of information on lightweight structural components is detailed in Table 3.

### 2.2. Origin of Research

Regarding the origin of the studies published in the 2015–2020 period, the USA leads the number of selected studies with 20%, followed by the UK and China (17%), Australia (10%), Germany, Italy, France (7%), and India, South Africa, Japan, Switzerland, and Spain (3%).

## 3. Trend Analysis in Structural Lightweight Materials

### 3.1. Materials

Based on the data analyzed, it can be concluded that, regarding structural lightweight materials, the materials that have raised the most interest during the last six years are aluminum alloys, titanium alloys, and steels, which appear in 30% of the studies each, followed by FRPs in 23.3% of studies. In analyzing the percentages, it is necessary to consider that an article may include several materials. In a 20-year analysis, the predominance of these materials is maintained, highlighting the importance of titanium alloys (27%) and aluminum alloys and FRPs (27%), followed by steels (24%), magnesium alloys (11%), and nickel alloys (8%).

#### 3.1.1. Aluminum Alloys

Aluminum alloys are among the favorite materials, from the engineering point of view, in applications of the automotive, aeronautical, and aerospace sectors, because of the excellent ratio of mass to mechanical and thermal properties [10]. During the period under analysis, aluminum has been studied in several lines of research, such as, for example, the joining between dissimilar materials in which one of the materials is an aluminum alloy. Multimaterial components allow the use of “energy-intensive” lightweight materials such as aluminum, since it favors their application only in areas where they provide a greater mass reduction.

Gullino et al. [12] conducted a review in 2019 aimed at the automotive sector of welding technologies applicable to aluminum-to-steel joining. The techniques reviewed include spot welding, continuous laser, electric arc, transition joint (TJ), magnetic pulse sheet (MPS), friction stir welding (FSW), friction bit joint (FSJ), and ultrasonic spot welding (USW). The paper concludes that the main risks of the aluminum-to-steel hybrid welded joint are the costs in a large-scale application and the potential corrosion in specific environments. Most of the scientific literature associated with the subject is mainly oriented to mechanical performance analysis. However, during the welding process, phases and compounds are formed, and it is unknown whether they affect corrosion.

Another of the research topics associated with aluminum is the analysis of residual stresses after a manufacturing process, particularly in parts of great responsibility and exposed to extreme conditions and fatigue, such as those of the aeronautical sector. In this sense, the quenching process is used to manufacture high-strength aluminum aerospace alloys, and it generates residual stresses because of the lack of uniformity during cooling. Forged parts need to reduce residual stresses from quenching by cold compression. Parts manufactured by machining can deform if residual stresses are not eliminated, causing a costly problem for manufacturers in the aerospace industry. The currently used process of cold compression relief reduces residual stresses but does not eliminate them. For Prime et al. [14], it is crucial to understand how these stresses are generated and how predictive techniques can mitigate them. For this purpose, in 2003, they proposed a study in which residual stresses in samples of two different aluminum alloys subjected to various treatments were measured and predicted by finite element simulation.

Moreover, aluminum matrix composites (AMCs) are composite materials with excellent properties: good mass-to-strength ratio, good ductility, high strength and modulus of elasticity, low coefficient of expansion, excellent wear and corrosion resistance, high creep temperature, and good fatigue behavior. For these reasons, they are widely used in the automotive and aerospace industries in applications such as robots, high-speed machinery, high-speed rotating shafts, automobile engines, and brake parts. The reinforcement microstructure of the matrix must be homogeneous, i.e., the ceramic particles must be homogeneously distributed in the metal matrix. The degree of reinforcement of the final multi-material depends on the amount, distribution, size, and shape of particles. The most commonly used particles in aeronautical structural applications are SiC and B4C [10]. In addition, the use of conductivity-enhancing particles within the aluminum matrix, such as TiB2, provides an increase in conductivity sufficient to be machined by non-conventional manufacturing processes such as electric discharge machining (EDM) [13,19]. This type of reinforced composite has been used, in recent years, in structural components with stringent requirements in the aerospace, defense, automotive, and sports sectors. Table 4 shows the studies included on aluminum alloys in the present study, listed by average number of citations and showing the main characteristics.

#### 3.1.2. Titanium Alloys

Titanium alloys offer good corrosion and fatigue properties and excellent mechanical characteristics, thanks to their metallographic structure. In addition, the increasing use of FRPs is pushing aircraft manufacturers to replace aluminum alloys with titanium alloys, because of the incompatibility between aluminum alloys and carbon [21]. However, titanium alloys are considered a challenging material to machine due to their low thermal conductivity, low modulus of elasticity, and high chemical affinity with tool materials [16]. According to the average number of article citations per years published, the most popular article is focused on titanium alloys, with 75.8 citations/years published, and its topic is additive manufacturing using welding. Williams et al. [21] conducted a study in 2016 in additive manufacturing using WAAM technology; a scheme is shown in Figure 2.

This process allows the manufacture of parts weighing more than 10 kg, with high deposition rates and low material and tooling costs. In addition, its good structural properties make it a candidate for manufacturing processes of medium–low complexity parts. It is used to manufacture titanium alloy parts for landing gear, wing reinforcements, and wind tunnel parts. The study shows strategies to deal with residual stress, the mechanical properties obtained, the removal of porosity, and the benefits of non-destructive tests on the parts obtained by this method. The key benefits of using this technology are the cost, the open architecture that allows the equipment to be customized, and the fact that there is no limitation on the size of the parts for materials that do not require a protective gas shield, such as aluminum alloys and steels. In 2018, Tabernero et al. [30] also carried out a study on WAAM technology applied to the titanium additive manufacturing process. The research was focused on a comparative analysis of the resulting mechanical properties obtained in experimental tests using three possible techniques, plasma arc welding (PAW), gas tungsten arc welding (GTAW or TIG), and gas metal arc welding (GMAW or MIG). The principal conclusions show the importance of the choice of the correct manufacturing parameters. Additionally, the GMAW process seems more suitable for large stainless-steel parts with low to medium mechanical requirements. In contrast, GTAW seems more adapted to medium and small sizes of titanium alloy and stainless-steel parts with medium and high mechanical requirements, and PAW is recommended for medium and large parts with medium and high mechanical requirements.

On the other hand, two articles focused on additive manufacturing using the SLM process were also published. Uhlmann et al. [32] analyses the characteristics and mechanical properties obtained in titanium alloys pieces manufactured by SLM. The analysis includes comparing properties such as density, microhardness, surface roughness, strength, and fatigue behavior compared to cast parts. The resulting specimens undergo post-treatment by hot isostatic pressing (HIP) heat treatment, which improves surface quality, seals micro-cracks, and reduces surface stresses. Computed tomography analysis reveals the effectiveness of the post-treatment, with porosity in untreated HIP specimens of 3.05% (relative density of 96.95%) and 0.81% (relative density of 99.15%) in post-treated samples, with a pore volume reduction of more than 160%.

Furthermore, the aerospace industry demands high quality and predictability of the materials that make up its equipment, both at static and dynamic loads. There is little literature on composites manufactured by laser metal fusion (LAN) technology in which fatigue behavior is analyzed, particularly at very high cycles. For this reason, Wycisk et al. [34] propose an article where the fatigue properties of UNS R56400 parts, additively manufactured using the LAN process and tested under stress-stress cycles of 107 cycles and stress-compression cycles at 109 cycles, are analyzed. The fatigue crack initiation is different in tests with different numbers of cycles: for tension–tension cycles with Nf < 106, the crack initiation occurs on the surface, for tension–tension cycles with Nf > 106, the crack initiation occurs in a defect inherent to the additive manufacturing inside the specimen, and for tension–compression cycles with Nf > 107, the cracks initiate in internal defects and there is no apparent influence of the frequency on the fatigue life. Depending on the size and location of the defects, the fatigue strength can vary significantly. Similarly to the Wycisk study [34], hot HIP post-treatment eliminates the inherent defects in the process and greatly improves fatigue performance. The results do not show a fatigue limit, but fatigue life decreases as the applied load increases.

Regarding studies aimed at optimizing machining processes, in the article ranked second in terms of the annual average number of citations, in 2015, Li et al. [63] studied the improvement of the milling process for thin-walled titanium tubes. The aim was to avoid vibrations resulting from interference between the tool flank and the surface to be machined. These vibrations produce marks in the machining that may require subsequent repair or cause tool breakage. In most studies, the effect of vibration is ignored because it is challenging to model. This study develops a non-linear dynamic model that takes into account the vibration during the milling of a thin-walled titanium alloy part, and despite being a simplification that only considers sharp edges, it allows the design, according to the model, of an anti-vibration clearance angle that manages to eliminate vibration and improve surface quality.

In addition, and continuing with the optimization of the machining process, but focused on the drilling process, in 2017, Xu and El Mansori [22] performed an experimental study on the wear characteristics of polycrystalline diamond (PCD) drills during orthogonal cutting of stacked CFRP/Ti hybrid composites. Two strategies for the cutting sequence Ti → CFRP and CFRP → Ti and different fiber orientation angles are used in the experimental test. The authors conclude that brittle fracture and elastoplastic deformation occur in the drilling process of CFRP/Ti stacks. Furthermore, the cutting sequence significant influences on the chip behavior and the final machining surface quality. The variation of forces that occur at the interface between materials, and the fragility of the PDC itself, are the origin of cracks on the edges. The orientation of the fibers modifies the length of the tool-part contact, affecting wear. Additionally, the significant factors identified on the surface roughness were the feed rate (f) and the fiber orientation (θ).

Otherwise, regarding the turning process of titanium alloys, some literature has also been published on optimizing process parameters and conditions and integrating sustainable cooling in the machining process. Ti-5553 alloy has high strength and excellent properties at both ambient and high working temperatures. It has better corrosion and fatigue resistance than UNS R56400 alloy, which makes it an ideal substitute. This alloy has already been used in the aircraft industry in components for landing gear, fuselages, wings, corrosion-prone or difficult to inspect areas, and in the military sector. However, this alloy has high reactivity leading to rapid tool wear, low thermal conductivity leading to high temperature, and heat band formation leading to high dynamic loads and vibration during machining. In the article [23], Sun et al. report a study on machining forces, finishing surfaces, and tool wear for Ti-5553 alloy, using cryogenic cooling, and comparing the results with those of flood cooling and minimum quantity lubrication (MQL). In the experimental trial, the machining forces are decreased when using cryogenic cooling up to 30% compared to MQL. However, MQL cooling achieves a better finish, as it is more ductile at higher temperatures. On the other hand, cryogenic cooling shows less wear of the insert nose. The study also presents a finite element model that simulates the cutting forces in cryogenic machining, with a good correlation between experimental and predicted results. In the comparison carried out in the study, abrasive wear is found for all three types of cooling. The forces are lower when using cryogenic cooling, with lower adhesion and lower temperature in the cutting zone resulting in less adhesion of material on the insert. The reduction in machining force is lower when using a lower feed rate, as the temperature increases less and the plastic deformations are smaller. On the other hand, surface roughness increases with increasing feed rate for the three types of cooling systems compared, cryogenic, MQL, and flooding. Table 5 shows the studies included on titanium alloys in the current study, listed by the average number of citations, and shows their main characteristics.

#### 3.1.3. Fiber Reinforced Polymers

Fiber-reinforced polymers (FRPs) have attractive properties as lightweight structural materials. These include stiffness, mechanical strength, and corrosion resistance. The most important families of FRPs are CFRP, glass fiber reinforced polymer (GFRP), and fiber metal laminates (FML). Fibers provide lightness, stiffness, and strength, and matrix provides load-bearing capacity and structural integrity. The mechanical properties of the composite material depend on the layering and orientation of the fibers. Maximum strength is obtained in the fiber direction and minimum strength in the perpendicular direction. FRPs show global characteristics of anisotropy and brittleness that make machining difficult [16]. CFRPs have attractive properties such as low density, high stiffness-to-mass ratio, excellent fatigue behavior and wear resistance, high dimensional stability, and low coefficients of friction, expansion, and electrical conductivity [65].

CFRPs have a remarkable role among the aircraft materials manufactured by Airbus, as they are lighter than aluminum, more mechanically resistant than iron, and more corrosion resistant than both, and provide a mass reduction fundamental to fuel economy. The use of this composite material in the manufacturing of the A350 XWB model has been even more extensive than in previous models; for example, most of the A350 XWB’s wing is composed of CFRPs, including its upper and lower covers. At 32 m long by 6 m wide, it is one of the largest aircraft components ever made of this material [66]. The first studies on CFRPs were carried out in 1980, and it is one of the most anisotropic materials, since its longitudinal tensile strength is greater than 5000 MPa while its shear strength is minimal [67]. However, it presents some disadvantages, such as the difficulty of machining originating from its abrasive nature and the low thermal conductivity.

Machining operations can reduce CFRPs’ structural strength and affect fatigue behavior. Drilling is the major machining operation because of its importance for the structural assembly of aircraft parts, and the critical challenge is to reduce the risk of delamination. Agius et al. [67] performed a study in 2015 in which they took into account fiber orientation and material anisotropy for the analysis of the thrust force generated during drilling. Initially, they investigate the exit damage produced during drilling under a variety of configurations. Subsequently, they propose a model that analyzes the thrust force concerning the generated load for each angular position. Finally, they correlate the thrust force and the exit defect in the drill hole. For the experimental trial, they use a specimen with multidirectional reinforcement fibers, similar to the components used in the aerospace industry, and the second specimen of unidirectional reinforcement fibers designed specifically for the experiment. The damage generated by the drilling process in CFRP is not homogeneous around the perimeter. As shown in Figure 3, it is characterized by a maximum ring diameter and a zone of angular delamination. Furthermore, the delamination depends significantly on the angle between the feed force and the fiber direction. In this case, this angle constantly changes during the rotation of the drill bit.

On the other hand, tool geometry has a significant effect, and the tip angle is considered by the authors to be the most important parameter to avoid delamination, since no delamination is found when the drill tip angle is smaller than 110°. The trial concludes that the best results are obtained for high cutting speeds, low feed rates, and tip angles less than 110°. On the same line of thrust force analysis, in 2017, Ojo et al. [68] proposed an analytical model of critical thrust force in CFRP drilling, based on a variation of the first-order shear deformation theory (FSDT) that considers the effect of shear deformation in the calculation of the critical thrust force. The delamination effect is shown in Figure 4a.

In 2012, Karpat and his colleagues conducted an investigation [69] on CFRP laminates of fabric material to investigate the influence of the geometry of a drill bit with a double angle on the tip through an experimental test on aerospace-grade fabric material. The research examines the effect of tip angle and drill bit coating on the cutting forces, torques, tool wear, and hole quality (dimensional, positioning, and surface quality). Figure 4b shows that the relationship between thrust force and the number of holes drilled at different rotational speeds is presented in a graph. The research concludes that higher feed rates result in higher bit wear and that feed rate has more influence than cutting speed. Hole diameter variation due to bit wear is also monitored, finding that, at high feed rates, diameter tolerance is more problematic than delamination in this type of fabric CFRP composite. In diamond-coated drills, fracture of the diamond layer significantly increases the thrust forces and, consequently, the appearance of delamination. On the other hand, the properties of the CFRP composite, the geometry of the drill bit, and the rigidity of the processing machine influence the process. Furthermore, the drilling process of FRPs produces an amount of heat that can cause the resin temperature to rise above its threshold temperature, resulting in thermal degradation. For epoxies, this critical temperature is in the range of 120 °C to 270 °C. These phenomena are more severe in dry machining processes employed in the aeronautical industry, where the cooling fluid appears to be detrimental to the maintenance of the chemical–physical properties of the resins of the composite materials. This machining difficulty frequently generates defects, such as fragmentation or thermal damage of the resin, and reduces the fatigue resistance of the material.

In parallel to drilling, other innovative machining processes are also being investigated. For example, laser machining of CFRP is challenging due to the inhomogeneity of the material. The transversal excited atmospheric pressure (TEA) technology employs an eight-microsecond short-pulse gas laser that works with a mixture of CO_2_, N_2_, and H_2_ at high pressures and achieves high peak power in very short pulses up to 250 kW, and is thus able to reduce heat damage. TEA technology is already utilized in fields such as surgery, laser processing of materials, or non-destructive testing. Salama et al. [17] make an interesting experimental test on cutting a CFRP using this type of laser. Figure 5 shows a transversal section of the result of the drilling performed by using the optimal parameters found during the experimental test. In the study, a design of experiment (DoE) is planned, and data obtained are analyzed using an analysis of variance (ANOVA). The factors initially identified as possible influential factors are laser fluence, repetition rate, and scanning speed. The response parameters on which their effect has been analyzed are heat affected zone (HAZ) size, depth of cut, and material removal rate (MRR). Based on the study, an optimization of process parameters is also conducted to minimize HAZ and maximize MRR. The ANOVA for HAZ shows laser fluence and scanning speed as the most significant factors. The ANOVA for machining depth shows laser fluence, repetition rate, and scanning speed as significant parameters. The ANOVA for MRR indicates that repetition rate and laser fluence are the parameters with influence. Moreover, there are significant interactions for all of them.

Another innovative machining process applied to CFRP is rotary ultrasonic machining (RUM), a hybrid grinding process aided by ultrasonic vibration of the order of 20 kHz, in which the coolant removes the chips and prevents overheating. Ning et al. [65] compare drilling of CFRP parts by conventional twist drill method and RUM. The cutting force, torque, surface finish, hole diameter, and material removal rate are analyzed in this work. The cutting forces found are consistently lower for RUM, although the difference is smaller at higher speeds. Also, for higher rotational speeds, the cutting force, torque, surface roughness, and hole diameter are lower. There are no significant differences in MRR between the two technologies. One of the drawbacks of RUM is the need for a significant initial investment in machinery and tooling, but the cost of the tool is cheaper because of reduced tool wear. The study’s overall conclusion indicates a higher efficiency of the RUM process, including better cutting forces, lower torques, and better surface qualities. Figure 6 shows the schematic of a state-of-the-art ultrasonic RUM machining process applied to FRPs.

Pecat et al. [20] conducted a study of the circumferential milling process of a unidirectional CFRP composite. The cutting mechanism is different for each fiber orientation. The damage also depends on the temperature of the workpiece to be machined; at lower temperatures, the machining forces are higher, but at higher temperatures, the defect caused by fiber bending is higher. The paper concludes that the optimum temperature zone to avoid both problems in the case of CFRP milling is around 80 °C. In addition, high cutting speeds lead to fiber bending defects in the subsurface regions.

On the other hand, CFRP/UNS R56400 and GFRP/UNS R56400 configurations are common in the aerospace industry because of their high specific strength and high corrosion resistance. They are considered an ideal alternative to conventional metals. The machining of hybrid composites in a single operation is challenging due to the different characteristics of the materials, which leads to different optimum machining conditions, aggravating the difficulties at the interface between the materials. FRP is highly anisotropic, with fiber reinforcement and low thermal conductivity, while polymer has ductile behavior. Titanium alloy exhibits low thermal conductivity, low elastic modulus, and high chemical affinity. Xu et al. [16] performed a detailed state of the art in 2016 regarding the advances in FRP/Ti hybrid composite drilling, paying particular attention to MRR and tool wear. Mechanical fastening by rivets or fasteners is very common in joining hybrid composites because it is a reliable joint, facilitating assembly and future inspection or maintenance operations. For this joint, the drilling operation is often performed in a single process to improve positioning tolerances. Typical drilling damages in FRP are crater in the matrix, interlayer delamination, fraying, and thermal damage. The most frequent defects in titanium are errors in hole size, roundness, position, and the existence of burrs. Titanium chips can cause scratches at the interface and in the CFRP, and the number of holes drilled by the tool are affected by tool wear. Heat is also challenging to dissipate at the interface of the two materials, causing degradation of the FRP.

In the same direction, Kumar et al. [29] report a study on the drilling of a CFRP/UNS R56400 composite in which they preliminarily identify tool wear due to the anisotropy of CFRP and the tendency to produce a built-up edge (BUE) of titanium as a possible difficulty. The machining of two materials with such different properties requires a compromise between tool geometry and cutting parameters. This often results in severe tool wear, increased cutting forces, poor hole quality, or large burrs. Excessive heat from machining titanium produces numerous quality problems in CFRP.

In the study, 100 bores are drilled at 118° and 130° drill angles, and tool wear is measured by microscope every ten drills. It is concluded that the number of drills made has a clear impact, and the best results are obtained with 130° drills, as shown in Figure 7. Abrasion is the wear mechanism that has the most effect on flank wear. Chisel edge wear depends on feed rate and relief angle, increasing with feed rate and small relief angles.

Finally, there is also literature on the development of new equipment to increase the flexibility and quality of FRP manufacturing processes. Automated tape laying (ATL) and automated fiber placement (AFP) are two processes currently employed in aircraft wing skin and fuselage applications. New developments are moving towards the use of robots to increase production flexibility and reduce equipment size. AFP heads are currently highly integrated into the machine and are not easy to replace when they become obsolete. Denkena et al. [62] present preliminary results of an AFP-type research prototype for automated fiber placement in thermoset CFRP. The study aims to enable the fabrication of rigid structures with complex curves for the aerospace industry. Table 6 shows the studies included on FRP composites in the present research, listed by the number of citations and their main characteristics.

#### 3.1.4. Special Alloys

The development of a new alloy is a long-term process; for example, in the case of gamma alloys, the first commercial flight equipped with UNS L54822 low-pressure turbine blades (LPTBs) (General Electric, Fairfield, CA, USA) engines was in March 2012, but before that point, the alloy had been used two years earlier for the first time in flight, it had been successfully deployed eight years earlier for the first time in GEnx engines, it had been certified 17 years earlier for the first time in CFMCF6 engines (General Electric, Fairfield, CA, USA), the alloy 4822 had been developed 24 years earlier, and the first exploratory research on gamma alloys had begun no less than 40 years earlier. In 2018, Kim and Kim [25] conducted a historical review of gamma alloys and their in-service temperature limitations and described strategies to develop gamma alloys with higher operating temperatures through the application of specific research and development plans.

Another group of alloys with exciting properties are the Ni–Ti alloys. Their differentiating properties are memory effect and superelasticity, but they also have low rigidity, biocompatibility, damping capacity, and corrosion resistance. However, their manufacture by conventional processes is challenging. For this reason, Elahinia et al. [26] conducted a state of the art review of the potential manufacturing techniques in 2016.

The main traditional manufacturing techniques are casting, which works at high temperatures, degrading functional properties, with a need for finish machining and excessive tool wear, and powder metallurgy (PM), which allows for the production of close to net shapes but has the disadvantage of a high level of impurities, with a process limitation on the control of form and porosity in complex parts. The study aims to advance towards an additive manufacturing method to produce intricate shapes of potential use in the aeronautics and biomedical industries. There are three key points when designing a manufacturing process for this alloy: the first is preparing the Ni–Ti powder, where the ratio of the elements is crucial for the final properties obtained. The second is the choice of process parameters that will affect density and impurities and, thus, the final properties. Finally, the third is the use of an inert atmosphere to minimize oxidation and impurities and improve the surface quality and density. The reduction of impurities is essential in biomedical industry applications. The most frequent problems in additive manufacturing are structural defects, pores, cracks, and residual stress. After a good choice of process parameters, the functional behavior is similar to conventional manufacturing. However, a heat treatment is needed to create the precipitates that provide the superelasticity, and then the oxidation caused needs to be removed by polishing or machining. For this reason, and because of the current limitations in machining and polishing complex parts, it is not possible to obtain the superelasticity feature in intricate geometries. Finally, there are also bibliographical references to nickel-, aluminum-, and cobalt-based superalloys. Two-thirds of these alloys are used in aeronautics to manufacture jet engines, and the remaining one third is used in the chemical, medical, and structural industries. These superalloys have unique properties of high-temperature resistance, hardness, and resistance to wear and corrosion. This fact is functionally very positive, but makes them very difficult to machine, as they have a low thermal conductivity that causes the heat to concentrate in the tool/workpiece and tool/chip contact zone, producing very high cutting temperatures and accelerating tool wear. In 2004, Ezugwu [28] carried out a state of the art of technologies developed during the preceding decade and applied to the machining of superalloys: self-propelled rotary tooling technology (SPRT), in which both the tool and the part to be machined are equipped with rotational movement, thereby reducing the tool–part contact time and increasing its useful life; high pressure coolant supply (HPCS), which introduces high-pressure coolant into the cutting zone to reduce its temperature in materials such as Inconel 718, and through which tool life was increased seven times; or more well-known cooling/lubrication strategies such as MQL, which is based on the use of a minimum amount of water-soluble oil, supplied to the cutting edge by compressed air, or cryogenic cooling (CC), that cools far below the softening temperature of the tool.

#### 3.1.5. Magnesium

Magnesium is a metal with many advantages regarding its use. It represents 2.7% of the earth’s crust and can be produced from seawater with a purity of 98.8%. Its density is 66% of that of aluminum and 25% of steel, making it an ideal candidate to replace them. Magnesium is the lightest structural alloy on the planet, but its low workability limits its use compared to other light metals such as aluminum. In pure laminated polycrystalline magnesium, cracks and fractures occur when the thickness is reduced by approximately 30%. One of the possible lines of work to improve this is the use of alloying materials, but no super-formable magnesium alloy has been developed, so far, at room temperature. On the other hand, if the grain size is reduced to microscopic dimensions by severe plastic deformation, grain boundary slips can be activated to improve ductility. Along these lines, Zeng et al. [70] carried out a study in 2017 of a super-formable magnesium alloy at room temperature without alloying and in which no cracks are produced during forming. The alloy is obtained by extrusion at 80 °C. In conventional extrusions at 150–400 °C, cracks and fractures occur when the thickness is reduced by compression by 20–30%. However, if the extrusion is carried out at 80 °C, the thickness can be reduced from 10 mm to 1.5 mm without fracture and with a bending capacity far superior to the conventional process. Figure 8a shows the result of a room temperature compression test with specimens made by both methods. The reduction in the grain size originates this superductility. The grain size goes from 82 μm in the alloy extruded at 400 °C without compression, to 1.3 μm in the alloy extruded at 80 °C. When the grain size is reduced to these levels, intergranular sliding along the grain boundaries can be activated at room temperature. Furthermore, this discovery is applicable to other metals with hexagonal structure.

Avvari et al. [72] also seek to improve the mechanical properties through grain size reduction by severe plastic deformation (SPD), and the specific process of equal chamfer angular pressing (ECAP) is part of this technique. The work evaluates the grain size obtained in a wrought UNS M11311 alloy subjected to a 4-step ECAP at 573 K. The grain sizes and mechanical properties are measured before and after each step and analyzed. However, the work seems inconclusive, as the mechanical properties and ductility of the material increased after the first two ECAP steps but subsequently decreased in the third and fourth ECAP steps, and decreased the tensile strength with the number of ECAP steps applied was also observed.

In another work related to this line of research, Liu et al. [15] carried out a study in 2019 to understand the parameters influencing ductility in magnesium alloys and their relationship with the mobility of pyramidal dislocations. The work proposes strategies to improve the ductility of magnesium alloys at room temperature. The ductility of magnesium alloys is strongly related to one type of pyramidal dislocations ⟨c + a⟩. The researchers aim to improve ductility by stabilizing these dislocations, preventing them from transforming into other structures. The findings of this study may also help to better understand other metals with a hexagonal structure. In this search for solutions to prevent the occurrence and propagation of cracks, thus improving their fatigue behavior, Barry et al. [71] propose using shot peening techniques to generate compressive stresses in the near-surface area. Shot peening enhances fatigue life by introducing compressive stresses, surface hardening, and increasing dislocation density. The research group investigates the fatigue life of sand-cast magnesium alloy A8 by characterizing its behavior before and after shot blasting treatment. The analysis of the tests allows positive conclusions to be drawn. Shot-blasted specimens offer a significant improvement in fatigue life, with up to five times increase in fatigue life depending on the amount of stress applied and a 30% increase in the fatigue strength limit of the material. In addition, compressive residual stresses of 100 MPa are found at a depth of 100 μm, the residual stress profile does not evolve significantly with fatigue loading, and, although the shot peening process increases surface roughness, crack initiation occurs mostly in residual pores rather than in defects caused by shot peening. Figure 8b shows a graph of stress amplitude versus number of cycles to failure for the blasted and non-blasted specimens. Table 7 shows the studies included on magnesium alloys and special alloys in the current study listed by number of averaged citations and shows their main characteristics.

### 3.2. Topics of Interest

Similar to the analysis of the materials studied, a survey can involve several themes in its development. The topics found in a larger percentage of the selected studies are considered trending topics. Regarding them, research that includes analysis of hybrid or multi-material component is reported in 30% of the selected studies over the last six years. Only 14% contained this topic in the period from 2000 to 2014. Within the studies that include hybrid components, the combination of materials most researched is metal + metal, reported in 44.5% of the studies on multi-materials in the period from 2015 to 2020. Table 8 shows the studies included on hybrid or multi-material components in the current paper, listed by the number of average citations and showing their main characteristics.

One topic that stands out in studies on multi-materials is the joining of dissimilar materials. The main methods of joining are mechanical, by drilling and subsequent bolting or riveting [16], and thermal, by different welding technologies [60]. In the case of drilling, the machining of multi-material composites in a single step is challenging, due to the different characteristics of each material that make them have different optimal machining conditions. This fact requires the use of a compromise between tool geometry and cutting parameters, and often results in severe tool wear, increased cutting forces, poor hole quality, or large burrs, leading to difficulties at the interface between materials [16,60]. For example, the most frequent defects in the drilling of titanium in hybrid composites are errors in hole size, roundness, position, and the existence of burrs. Titanium chips can cause scratches at the interface between materials and in the materials at the chip exit, and the number of holes drilled by the tool is also influenced by the effect of tool wear [16].

Regarding the option of mechanical joining of dissimilar material by riveting, Li et al. [59] carried out a state of the art review of the self-piercing riveting (SPR) technique in 2017. It is a process of mechanically joining two or more sheets of material by riveting through the top and bottom panels, with mechanical fastening on the last one. Figure 9 shows the main areas of an SPR joint and its possible defects. This method is currently used for aluminum and hybrid composite joints at leading automotive companies such as Audi, Jaguar Land Rover, Volvo, BMW, Daimler, Tesla, and Ford. Resistance spot welding (RSW) is used as an alternative technology. According to the authors, this process has clear advantages over other methods, as it is a sustainable and clean process, capable of efficiently joining different materials such as aluminum, steel, magnesium, copper, plastics, wood, or composites, does not require pre-drilling or heat treatments, can be combined with adhesives and lubricants, is low energy, has long tool life, low work cycles, airtight joints, no effect on the possible heat treatments of the material to be joined, and high static and fatigue resistance.

In addition, the following disadvantages are identified: access is necessary from both sides, there is a deformation on one of the sides, there is a possibility of galvanic corrosion, a high insertion force is needed for the rivet, and its use is limited in fragile materials. In the case of a brittle material, brittle material would be placed in the first layer, and the ductile material would be placed in the last layer where the mechanical fastening by deformation is performed. This method allows a more precise alignment of the components to be joined compared to conventional methods. When used in combination with adhesives, it increases the rigidity of the joints. The study concludes that SPR joints show a higher fatigue strength than RSW joints. The failure modes identified during the static tests are the tear strength of the last layer material, the mechanical strength of the top layer material and the rivet, the rivet interlocking distance in the last layer, the friction between faces, and the configuration of the stacked materials. The sources of fatigue failure vary, mainly occurring in the bottom layer material due to stress concentration from bending, while friction can also accelerate cracks. In measurements taken in the study, high residual stresses appear due to the mechanical deformations inherent to the riveting process. Protective surface coatings such as, e.g., e-coatings, adhesives, and paints are often applied to these types of joints to reduce corrosion.

On the other hand, regarding thermal joints, conventional welding techniques are founded on the fusion of the base material and the filler material along the welded joint. The cooling of the weld is the most critical part of this type of welding, and is where many of the defects occur, so it must be carried out in a protected atmosphere. In this sense, the FSW process has a high potential. Haghshenas and Gerlich [60] conducted a comprehensive state of the art of friction stir welding techniques in 2018, aimed at joining dissimilar materials and analyzing the factors that control the quality and strength of the joint. FSW does not require melting of the base materials, as the peak working temperature ranges from 0.6 to 0.95 of the melting temperature. It is one of the most widely used joining techniques in the automotive industry. Its main advantage is the low temperature in the welding zone, which prevents the formation and growth of undesirable intermetallic compounds that weaken the strength of the joint. In addition, it minimizes temperature deformations and residual stresses, has good tensile and fatigue properties, requires no consumables, and reduces environmental and health problems. A drawback of this process is that it is limited in the welding of large parts because of the high forces and frictions required. Also, the process involves control mechanisms to obtain reproducible and high-quality results. Deviations in thickness, temperature conditions, deformations of the clamping system, tool wear, or even an inaccurately positioned tool can lead to different weld qualities or defects [60,67]. The most commonly used response variable in studies characterizing these welds is ultimate tensile strength (UTS), grain size, and microhardness. The results show that FSW can achieve a UTS similar to that of the base material, and that it is not possible by traditional methods.

Finally, Gullino et al. [12] perform an interesting analysis of the state of the art of the different welding technologies available for joining dissimilar materials based on both melting of the base materials and solid joining technologies, focusing the analysis on the technologies applicable to joining steels and wrought aluminum alloys. The technologies examined include RSW, laser beam welding (LBW), arc welding, explosion welding and transition joints (EWTS), magnetic pulse welding (MPW), roll bonding (RB), FSW, friction bit joining (FBJ), and USW.

Furthermore, additive manufacturing is present in 26.7% of the selected studies in the past six years. No selected papers met the pre-established criteria in the previous period of 2000–2014. Table 9 shows the included studies on additive manufacturing listed by the number of averaged citations and shows their main characteristics. The most studied technique is SLM, an additive manufacturing process that manufactures parts layer by layer, using metal, ceramic, polymer, or composite powders as raw materials. At each stage, a substrate bed is deposited, and the deposited material is selectively melted using a laser beam. Each layer is formed according to information in a computer-aided design (CAD) format. The operation is repeated until the part acquires the final shape.

This process makes it possible to manufacture parts with high added value and complex shapes that are very difficult or impossible to manufacture by other methods [33]. One of the drawbacks of the technology is the creation of residual stresses that can limit the fatigue life of the parts. For this reason, several studies are being carried out to understand how they are generated, and procedures are being developed to reduce them by HIP post-treatment on UNS R56400 titanium parts [32,34], by controlling the layer deposition temperature [35] on UNS R56400 titanium, or by combining the SLM process with laser shock peening (LSP) [33]. LSP is a treatment that creates compressive residual stresses on the subsurface of the part and is used in applications with high-quality requirements, such as the aerospace and nuclear industries.

The effect of residual stress on fatigue life has been extensively studied, and compress residual stress (CRS) in the near-surface region has a positive impact. Furthermore, the deeper the CRS, the more near-surface cracks are mitigated, resulting in better fatigue resistance. The research group verifies the efficiency of a newly patented device by fabricating a UNS S31603 piece with compressive residual stresses in an experimental test, proving the device’s efficiency both in achieving significant increases in the magnitude of compressive residual stresses and in increasing their depth.

In addition to laser technology processes, other welding technologies such as WAAM have also been developed, reaching workpiece sizes of more than 10 kg in the titanium alloy UNS R56400 [21]. This technology offers important advantages, such as the possibility of creating multi-material compounds, its possible integration in hybrid machining centers that include additive manufacturing and conventional machining within the same manufacturing process, manufacture of parts with integrated functionalities, optimization of shapes and geometries, integration of circuits and internal cavities, manufacture of customized multi-materials, etc. One of its disadvantages is the considerable heat input that can produce deformations due to thermal stress. Still, it can be controlled using deposition strategies symmetrical to the plane of symmetry. Lockectt et al. published [31] a study in 2017 in which they compare the WAAM process with other types of additive manufacturing (AM) and establish a set of rules to take into account when designing manufacturing processes for parts to be manufactured using WAAM. The study also proposes a methodology for selecting the best process orientation based on the substrate material, deposited material, number of operations, build complexity, and symmetry. Although the method allows a complete fabrication process design to be achieved, the authors recommend reviewing it from an engineering point of view to ensure that a practical solution is reached. Finally, the authors illustrate the methodology with two examples of its application in the manufacture of aerospace parts. Furthermore, in an effort to improve competitiveness, reduce costs, and increase added value, hybrid machining centers that integrate additive manufacturing by laser deposition are being developed, evolving towards a done-in-one manufacturing process, where the aim is to produce close-to-net shapes that are finished on-site by high-precision machining. This process is particularly suitable for small batch production of difficult-to-machine materials where high precision is required. In 2016, Yamazaki [73] carried out a study on a hybrid machine tool model from the manufacturer Mazak, which includes examples of applications that have already been produced and potential future applications. As a use example, a workpiece for the oil industry with adequate mechanical strength is manufactured. The example employs stainless steel S31600 (316S31) as a substrate, a common material for pipes used in the marine oil industry due to its good corrosion resistance in seawater, and UNS N07718 metal powder is used as a deposition material, with very good high temperature, corrosion, oxidation, and creep resistance.

In pieces under cyclic loading and areas of stress concentration, in parts of additive manufacturing without post-treatment, cracks appear during ageing that can lead to fatigue failures, which is an important problem to avoid in the aeronautical industry. For this reason, 20% of the selected articles from 2015 to 2020 include the study of the fatigue behavior of materials in the research, and specific investigations are conducted on most of the materials, including aluminum [14], titanium [32,34], steels [33], magnesium [70,71], and special alloys such as Ni–Ti [26]. In addition, and to facilitate the early identification of this type of defect, technologies such as high frequency guided waves are developed to monitor the status of assembled and aged components. The aim is to easily and reliably locate possible hidden cracks in assembled multi-material components that are difficult to reach. In the study by Chan et al. [61], a multi-material component of two aluminum plates joined by an intermediate epoxy is analyzed. Crack growth in an anchor hole under cyclic loading is monitored using a laser interferometer which shows good sensitivity and repeatability for the location of small cracks. Figure 10 shows the most relevant parameters in a residual stress profile analysis.

On the other hand, 50% of the studies selected for both periods 2000–2014 and 2015–2020, including a machining study, survey the drilling process. This fact confirms the importance and interest, current and maintained over time, in developing the drilling process for the aeronautics and automotive sectors. Drilling plays a fundamental role in the assembly of parts in the aeronautical industry; it is estimated that it is used in 50% of machining operations. It is frequently used in structural components that require subsequent assembly using screws or rivets. During the drilling process of FRPs, a defect called delamination frequently occurs, reducing their strength and rigidity. This makes it a topic of great interest, either as a single composite [67,68,69] or as part of multi-materials [16,61]. The path followed by most works is the study of the critical thrust force, for which some works develop models taking into account the shear forces [68], and others build models depending on the geometry of the tool [29,67,69], or perform experimental trials and study the effect of influential factors such as the geometry of the drill bit and/or its coating [69].

In addition, the application of non-conventional manufacturing processes such as EDM [13], electrochemical machining (ECM) [19,74] and photochemical machining (PCM), photo electroforming (PEF), and laser beam machining (LBM) [64] to structural lightweight materials is also investigated. On the other hand, the machining processes of thin-walled parts are of great interest for the aeronautical and aerospace sectors to reduce the mass of aircraft and vehicles. Related to this topic, Li et al. [63] published an investigation in 2015 to reduce the vibrations produced during the milling of thin-walled titanium parts, which are due to interference between the flank of the tool and the surface to be machined. In the study, they propose a simplified model applicable to the process, which they subsequently apply to the design of a tool. They manage to reduce the vibrations and improve the surface quality. Many factors influence the creation of folds or wrinkles in thin-walled machining, such as stresses, the mechanical properties of the material, the geometry of the part, and, above all, the boundary conditions. These wrinkles can severely affect the functionality of the pieces and their lifetime properties, such as wear. For this reason, in 2015, Nan et al. [75] carried out a compilation of the main current prediction methods and summarized their principal advantages and limitations. The study proposes a new hybrid method that combines several of the existing methods compiled. They manage to predict plastic wrinkles mainly associated with complicated boundary conditions (CBC) processes. Table 10 shows the included studies on the drilling process listed by the number of average citations and shows their main characteristics.

## 4. Conclusions

For decades, the aeronautical and automotive industries have been pushing the development of new structural lightweight materials to reduce the mass of aircraft and vehicles in order to increase their energy efficiency, and at the same time to meet the stringent functional requirements demanded by these industries. On the other hand, the scope of structural lightweight materials is extensive. Therefore, it is considered helpful for other interested researchers to analyze the scientific literature published during the period of 2000–2020, paying particular attention to the recent trends defined by the period from 2015 to 2020.

The aim of the study is not only to carry out a state of the art through a narrative analysis of the scientific literature associated with the subject, but also to analyze and understand, through a systematic review, the trends and needs of researchers in the area of lightweight structural materials associated with studies in the aeronautical and automotive sectors.

Sometimes, in investigations based on existing studies or existing literature, not all publications, studies, or articles offer the same guarantees of veracity, methodological quality, and interest. For this reason, this study applies a methodology adapted from the PRISMA statement to engineering for limiting possible biases in the selection and analysis of the literature. The final objective is to have the most relevant and representative articles of proven quality, applying clear and homogeneous inclusion and exclusion criteria, and carrying out an unbiased analysis of the information obtained to conclude the current trends in materials and manufacturing processes used. The key points of the methodology are: prior definition of quality and inclusion criteria to select the literature, search using Boolean equations based on the previously established criteria, the definition of the search engine, an initial preselection, subsequent review to obtain the final selection, and, finally, unbiased analysis based on closed and univocal questions on the final selection of scientific literature.

Among the selection criteria applied, founded on previous sections, were: Open Access, English language, WoS as a search engine, annual average number of citations as a bibliometric indicator, and use of Boolean equations based on keywords. The objective is to know the current trend in the study of these materials by analyzing Open Access literature, and, in any case, to establish a ranking of publications. In a future step, the authors consider it interesting to carry out this same analysis including non-Open Access works and to compare results.

As a result of the systematic search, a final selection of 37 articles from the period of 2000–2020, 23 publications at Q1 (62%), 3 publications at Q2 (8%), and 11 publications at renowned conference proceedings (30%), with an overall average of 67.1 citations/(article x year published) has been obtained from a search in the WoS database on 3 July 2021. To facilitate the analysis of current trends and provide a longer time perspective, the global period of 2000–2020 has been divided into two periods: the recent period from 2015 to 2020 and the earlier period from 2000 to 2014. The final selection consists of the 30 articles with the highest average number of citations per year of publication during 2015–2020, and the 7 articles with the highest average annual number of citations from the period of 2000–2014.

Based on the data extracted and analyzed, it can be concluded that, regarding structural lightweight materials, the materials that have attracted the most attention during the last six years are aluminum alloys, titanium alloys, and steels, which appear in 30% of the studies each, followed by FRPs in 23.3% of studies (a study may contain several materials). In an overall analysis of the last 20 years, the relevance of these materials is maintained, highlighting the importance of titanium alloys (29%), aluminum alloys and FRPs (27%), followed by steels (24%), magnesium alloys (11%), and nickel alloys (8%). Regarding the topics identified as the most trending and relevant topics, 30% of the studies published over the last six years include a hybrid or multi-material compound. In contrast, only 14% included a hybrid component or multi-material compound in the period of 2000–2014. Among these multi-material studies, the combination of metal + metal is the most studied recent combination, with 44.5% during the period of 2015–2020. In the analysis of the percentages, it is necessary to consider that a study may include several materials and topics in its research.

Regarding the origin of the studies published in the period from 2015 to 2020, the USA leads the number of selected studies with 20%, followed by the UK and China (17%), Australia (10%), Germany, Italy, France (7%), and India, South Africa, Japan, Switzerland, and Spain (3%). In addition, all the information obtained is summarized in tables to facilitate the search and interpretation, by interested researchers, of the articles with the highest average number of citations per year published during the periods analyzed, and their main characteristics are shown as follows: aluminum alloys, titanium alloys, FRPs, special alloys, hybrid or multi-material components, additive manufacturing, and the drilling process.

## Figures and Tables

**Figure 1 materials-14-06577-f001:**
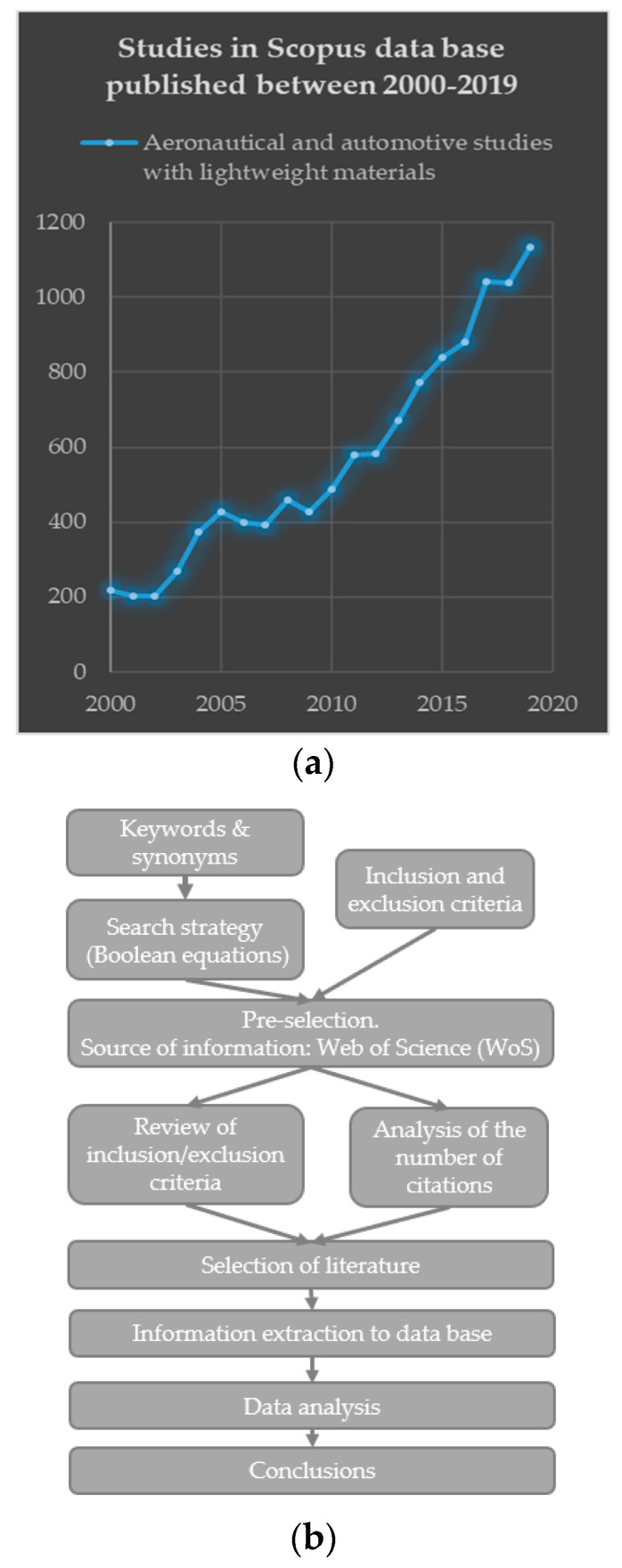
(**a**) Trend according to the number of studies on lightweight materials applied to the aeronautical and automotive sectors; (**b**) Defined strategy for literature search.

**Figure 2 materials-14-06577-f002:**
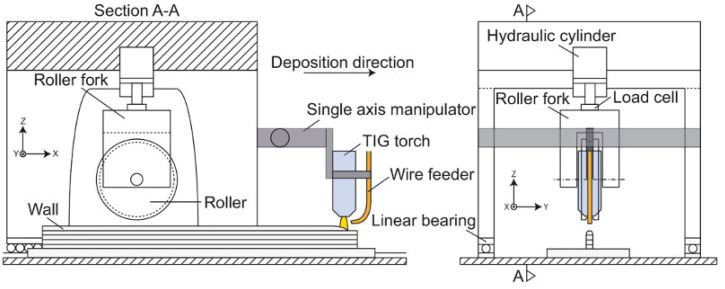
Schematic diagram of rolling and welding equipment [21].

**Figure 3 materials-14-06577-f003:**
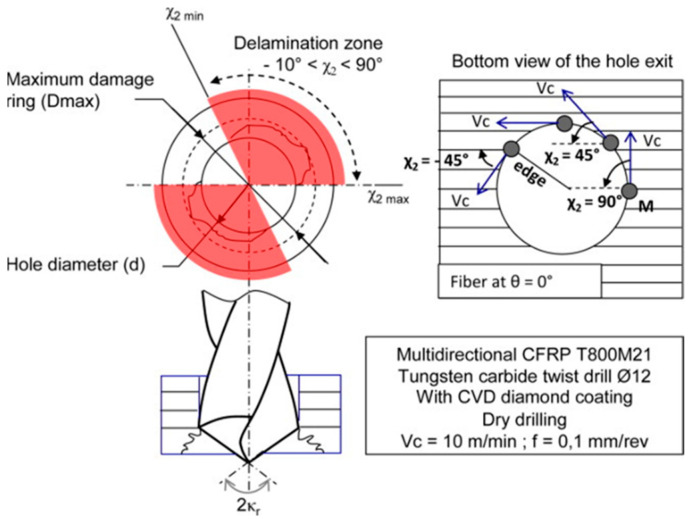
Definition of damage criteria during CFRP drilling [67].

**Figure 4 materials-14-06577-f004:**
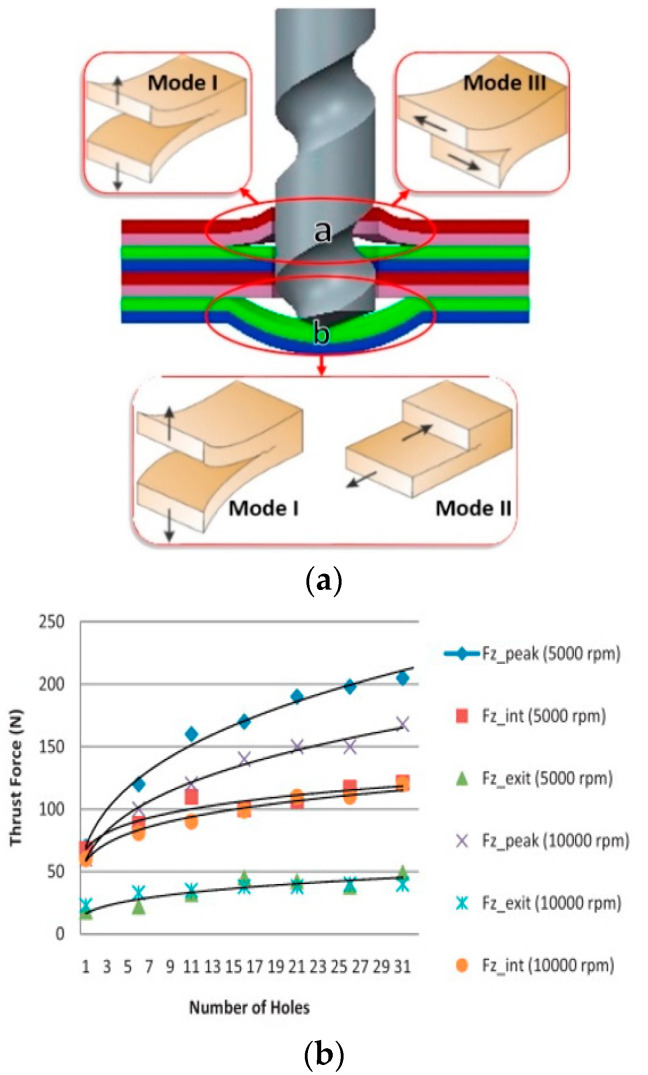
(**a**). Delamination phenomena depicting different modes during a-peel-up and b-push-out type [68]; (**b**) Rotational speed influence on thrust forces and holes number [69].

**Figure 5 materials-14-06577-f005:**
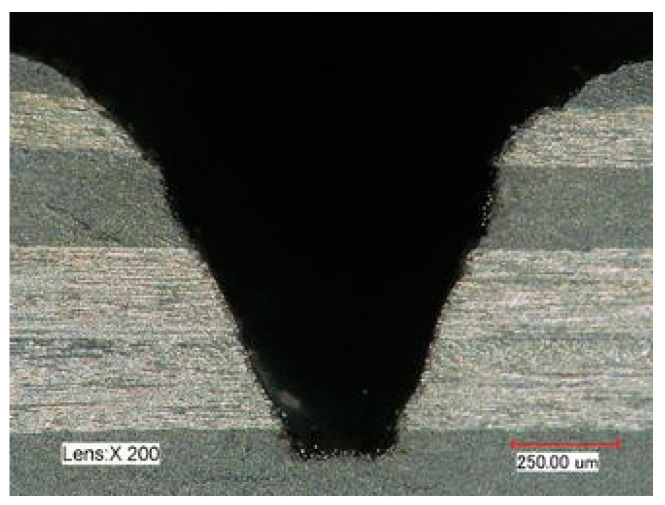
Cross-sectional view of the machined sample for optimum solution [17].

**Figure 6 materials-14-06577-f006:**
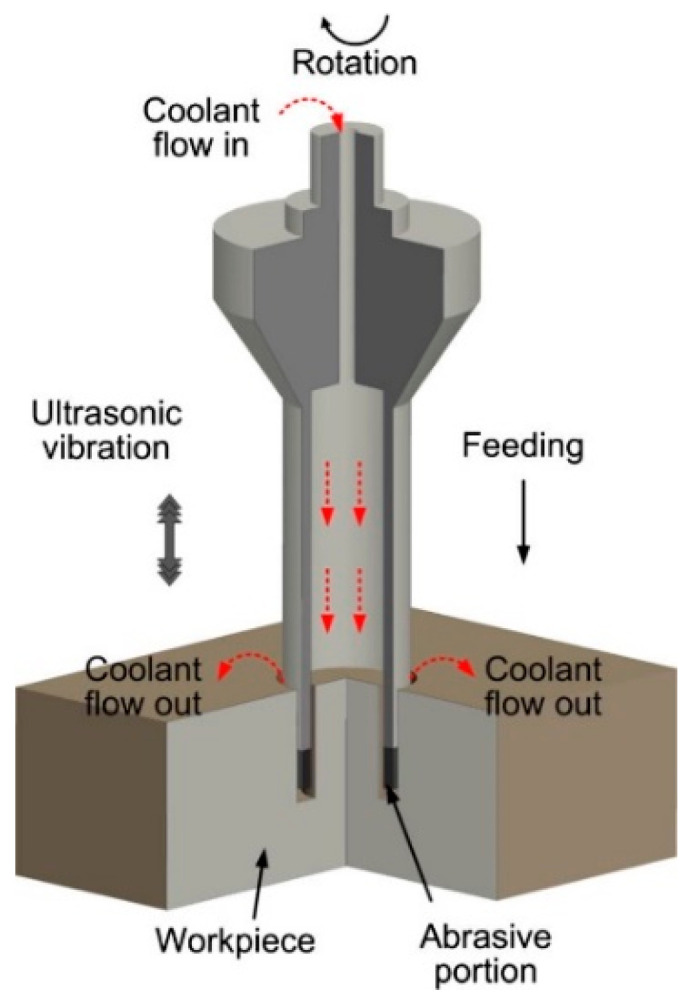
Illustration of rotary ultrasonic machining [65].

**Figure 7 materials-14-06577-f007:**
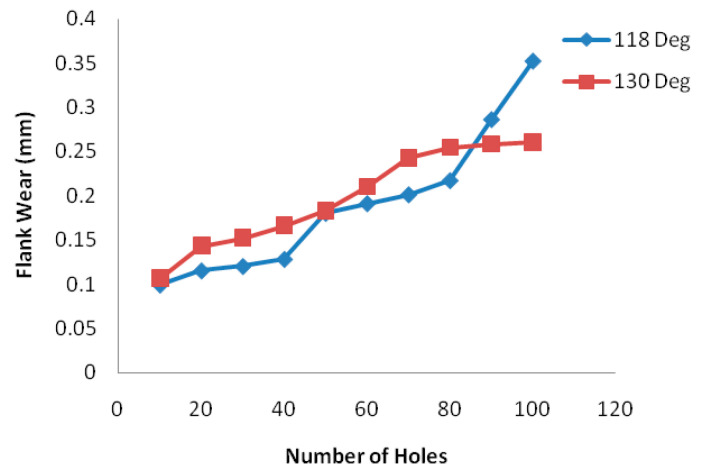
Effect of flank wear on point angle [29].

**Figure 8 materials-14-06577-f008:**
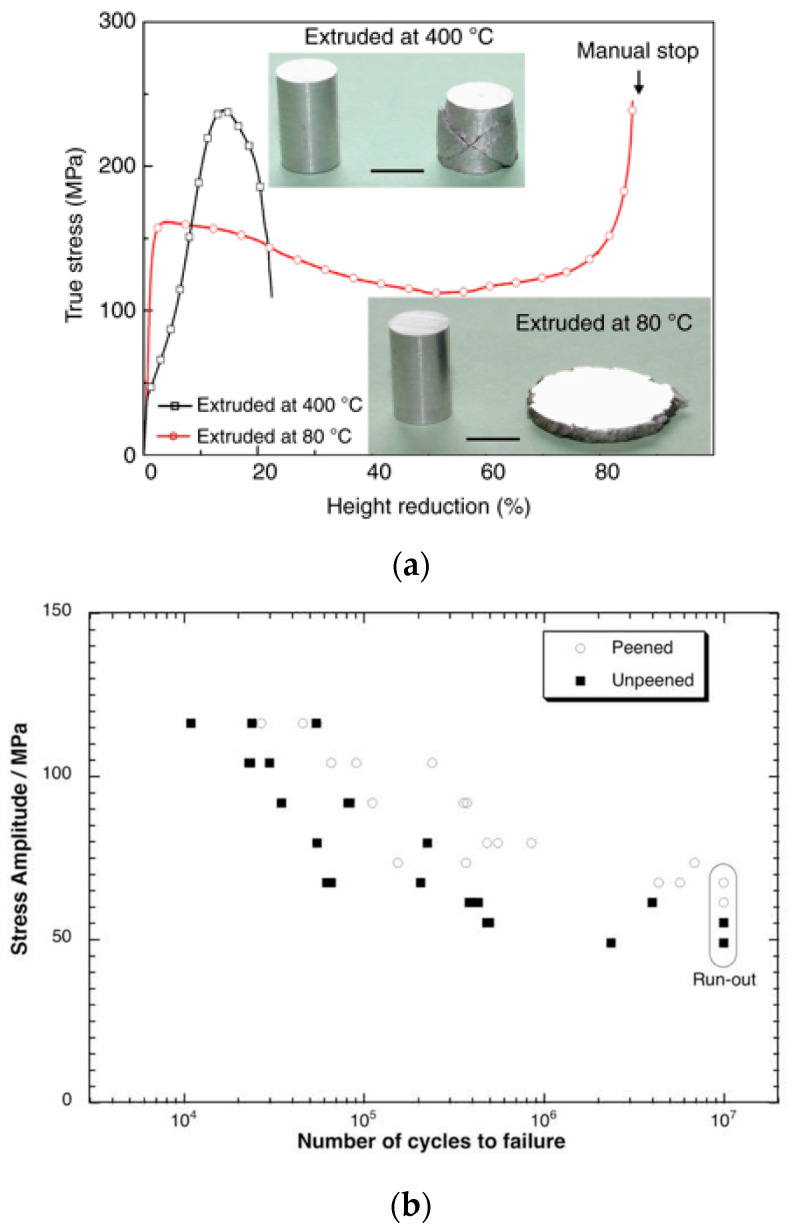
(**a**) Room temperature compression of specimens extruded at 80 and 400 °C with specimens before and after compression test [70]; (**b**) Stress amplitude versus number of cycles to failure for the peened and unpeened samples [71].

**Figure 9 materials-14-06577-f009:**
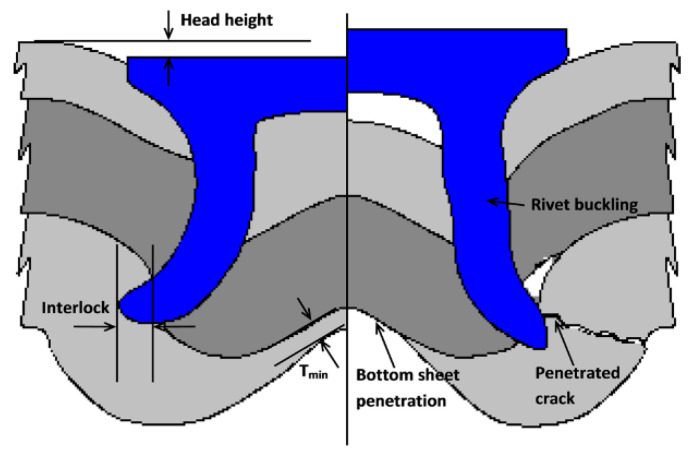
SPR joint quality and some faults [59].

**Figure 10 materials-14-06577-f010:**
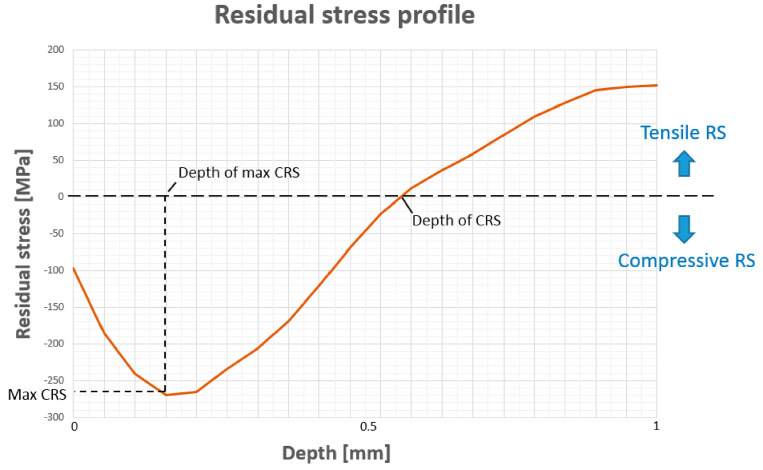
Residual stress profile displaying the most relevant parameters: Max CRS—maximum amount of CRS; Depth of max CRS—depth at which the maximum CRS is observed; Depth of CRS—depth at which a transition from CRS to tensile residual stress (TRS) occurs [33].

**Table 1 materials-14-06577-t001:** Average number of citations per article for each period, taking into account the number of years each individual article has been published.

Period	2000–2020	2015–2020	2000–2014
(Average number of citations)/(article x N° year published)	67.1	72.9	42.0

**Table 2 materials-14-06577-t002:** Main data associated with the quality of the scientific journals in which the selected articles have been published.

Period	2000–2020		2015–2020		2000–2014	
Articles selected	N°	%	N°	%	N°	%
Final selection	37		30	81.1%	7	18.9%
Proceedings Paper	11	30%	8	27%	3	43%
Q1	23	62%	19	63%	4	57%
Q2	3	8%	3	10%	0	0%

**Table 3 materials-14-06577-t003:** Selection criteria and main characteristics of the structural lightweight materials literature.

Criteria	Features
Publication period	6 years/2015–2020/30 articles most cited
	15 years/2000–2014/7 articles most cited
Date of last search	7 March 2021
Type of studies	Articles: Scientific journals Q1–Q2 and conference proceedings
Search strategy	Equation (1)
Bibliography sources	Only databases included in Web of Science
Language	English
Type of publication	Only Open Access

**Table 4 materials-14-06577-t004:** Summary of studies including aluminum alloys, listed by the average number of citations per years published.

Ref.	Citations Averaged	Publication Type	Year	Material(s)	Key Topics Addressed	Origin
[21]	75.8	Q1	2016	Al/Ti/Steel	^1^ AM	United Kingdom
[59]	12.75	Q1	2017	Al/Steel	^3^ SoA/^2^ HC	United Kingdom
[60]	11.67	* PP	2018	Al/Steel	^3^ SoA/^2^ HC/WT	USA
[10]	11.60	* PP	2016	Al/Ceramics	^3^ SoA/^2^ HC	South Africa
[12]	11.00	Q1	2019	Al/Steel	^3^ SoA/^2^ HC/^4^ WT	Italy
[61]	6.67	Q1	2015	Al/Polymer	^2^ HC	United Kingdom
[13]	6.00	Q1	2015	Al/Ceramics	^2^ HC/^5^ M	India
[31]	5.50	Q1	2017	Al/Ti/Steel	^1^ AM/^4^ WT	United Kingdom
[62]	5.00	* PP	2016	Al/FRP	Other(s)	Germany
[19]	4.75	* PP	2012	Al/Ceramics	^1^ AM	India

* Proceedings Paper; ^1^ Additive manufacturing; ^2^ Hybrid Component; ^3^ State of Art; ^4^ Welding Technology; ^5^ Machining.

**Table 5 materials-14-06577-t005:** Summary of studies including titanium alloys, listed by the average number of citations, taking into account the number of years the article has been published.

Ref.	Citations Averaged	Publication Type	Year	Material(s)	Key Topics Addressed	Origin
[21]	75.80	Q1	2016	Ti/Al/steel	^1^ AM	United Kingdom
[63]	50.67	Q2	2015	Ti	^5^ M	China
[16]	21.00	Q1	2016	Ti/FRP	^5^ M	France
[32]	16.00	* PP	2015	Ti	^1^ AM/^3^ SoA	Germany
[35]	16.00	Q2	2018	Ti	^1^ AM/^3^ SoA	Australia
[30]	8.67	* PP	2018	Ti/steel	^1^ AM/^4^ * WT	Spain
[23]	8.33	* PP	2015	Ti	^5^ M	USA
[22]	5.5	Q1	2017	Ti/FRP	^2^ HC/^5^ M	China
[31]	5.5	Q1	2017	Ti/Al/steel	^1^ AM/^3^ SoA	USA
[29]	4.23	* PP	2013	Ti/FRP	^5^ M	India
[64]	1.88	Q1	2008	Ti/Ni	^5^ M/^3^ SoA	China

* Proceedings Paper; ^1^ Additive manufacturing; ^2^ Hybrid Component; ^3^ State of Art; ^4^ Welding Technology; ^5^ Machining.

**Table 6 materials-14-06577-t006:** Summary of studies including FRP, listed by the average number of citations, taking into account the number of years the article has been published.

Ref.	Citations Averaged	Publication Type	Year	Material(s)	Key Topics Addressed	Origin
[16]	21.00	Q1	2016	FRP/Ti	^2^ SoA/^1^ HC/^3^ M	France
[16]	14.80	Q1	2016	FRP	^3^ M	USA
[65]	9.33	Q1	2015	FRP	^3^ M	France
[17]	7.20	Q2	2016	FRP	^3^ M	United Kingdom
[68]	6.00	Q1	2017	FRP	^3^ M	Italy
[22]	5.50	Q1	2017	FRP/Ti	^1^ HC/^3^ M	China
[69]	5.42	Q1	2012	FRP	^3^ M	Turkey
[62]	5.00	* PP	2016	FRP/Al	Al/Ceramics	Germany
[20]	4.33	* PP	2012	FRP	^3^ M	Germany
[29]	4.23	* PP	2013	FRP/Ti	^3^ M	India

* Proceedings Paper; ^1^ Hybrid Component; ^2^ State of Art; ^3^ Machining.

**Table 7 materials-14-06577-t007:** Summary of studies including magnesium and special alloys, listed by the average number of citations, taking into account the number of years the article has been published.

Ref.	Citations Averaged	Publication Type	Year	Material(s)	Key Topics Addressed	Origin
[26]	37.2	Q1	2016	NiTi	^1^ AM/^2^ SoA	USA
[25]	32.33	Q1	2018	Gamma alloy	^2^ SoA	USA
[15]	22.50	Q1	2019	Mg	Ductility enhancement	China
[70]	17.00	Q1	2017	Mg	Ductility enhancement	Australia
[71]	2.78	Q1	2009	Mg	Fatigue	United Kingdom
[72]	1.92	Q1	2013	Mg	Ductility enhancement	United Kingdom

^1^ Additive manufacturing; ^2^ State of Art.

**Table 8 materials-14-06577-t008:** Summary of studies including hybrid components, listed by the average number of citations, taking into account the number of years the article has been published.

Ref.	Citations Averaged	Type	Year	Hybrid Combination Type	Material(s)	Key Topics Addressed	Origin
[16]	21.00	Q1	2016	Metal + Polymer	FRP/Ti	^2^ SoA/^4^ M	France
[59]	12.75	Q1	2017	Metal + Metal	Al/Steel	^2^ SoA/^4^ M	United Kingdom
[60]	11.67	* PP	2018	Metal + Metal	Al/Steel	^2^ SoA/^3^ * WT/^1^ JDM	USA
[10]	11.60	* PP	2016	Metal + Ceramic	Al/Ceramics	^2^ SoA	South Africa
[12]	11.00	Q1	2019	Metal + Metal	Al/Steel	^3^ WT/^5^ JDM	Italy
[73]	8.40	* PP	2016	Metal + Metal	Ni/Steel	^1^ AM/^5^ JDM	Japan
[61]	6.67	Q1	2015	Metal + Polymer	Al/Polymer	Al/Ceramics	United Kingdom
[13]	6.00	Q1	2015	Metal + Ceramic	Al/Ceramics	^4^ M	India
[22]	5.50	Q1	2017	Polymer + Polymer	Ti/FRP	^4^ M	China
[19]	4.75	* PP	2012	Metal + Ceramic	Al/Ceramics	^4^ M	India

* Proceedings Paper; ^1^ Additive manufacturing; ^2^ State of Art; ^3^ Welding Technology; ^4^ Machining, ^5^ Joining Dissimilar Materials.

**Table 9 materials-14-06577-t009:** Summary of studies including additive manufacturing listed by number of citations, listed by the average number of citations, taking into account the number of years the article has been published.

Ref.	Citations Averaged	Publication Type	Year	Material(s)	Additive Technology	Key Topics Addressed	Origin
[21]	75.80	Q1	2016	Al/Ti/Steel	^4^ WAAM	^2^ WT	United Kingdom
[26]	37.20	Q1	2016	NiTi	^3^ SLM	^1^ SoA/Fatigue	USA
[32]	16.00	* PP	2015	Ti	^3^ SLM	Fatigue	Germany
[35]	16.00	Q2	2018	Ti	^3^ SLM	^1^ SoA	Australia
[33]	13.75	Q1	2017	Steel	^3^ SLM	Fatigue	Switzerland
[30]	8.67	* PP	2018	Ti/Steel	^4^ WAAM	Mechanical properties	Spain
[73]	8.40	* PP	2016	Ni/Steel	^3^ SLM	Hybrid manufacturing	Japan
[31]	5.50	Q1	2017	Al/Steel	^4^ WAAM	Design guide	United Kingdom

* Proceedings Paper; ^1^ State of Art; ^2^ Welding Technology; ^3^ Selective Laser Melting; ^4^ Wire Arc Additive Manufacturing.

**Table 10 materials-14-06577-t010:** Summary of studies including drilling process listed by number of citations, listed by average number of citations considering the number of years the article has been published.

Ref.	Citations Averaged	Publication Type	Year	Material(s)	Key Topics Addressed	Origin
[16]	21.00	Q1	2016	FRP/Ti	^1^ D/^2^ CTF	France
[65]	14.80	Q1	2016	FRP	^3^ RUM	USA
[67]	9.33	Q1	2015	FRP	^1^ D/^4^ A	France
[68]	6.00	Q1	2017	FRP	^1^ D/CTF	Italy
[22]	5.50	Q1	2017	FRP/Ti	^4^ A	China
[69]	5.42	Q1	2012	FRP	^1^ D	Turkey
[29]	4.23	* PP	2013	FRP/Ti	Metal + Polymer	India

* Proceedings Paper; ^1^ Delamination; ^2^ Critical Thrust Force; ^3^ Rotary Ultrasonic Machining; ^4^ Angle Force-Fiber.

## Data Availability

Data sharing not applicable.

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
