# Peer review of "Lightweight Structural Materials in Open Access: Latest Trends"

_materials, 2021, doi:10.3390/ma14216577_

Round 1

Reviewer 1 Report

The research topic is of great interest.

However, there are serious flaws in the methodology (procedures, presentation or analysis of the data).

The paper contains some observations based on a very limited literature (37 articles from the period 2000-20), which lead to poor conclusions (not relevant at all for scientists). The article is based on 30% articles in conference proceedings which most of the time are not per-reviewed.

Most important the present study ignore important work in the field, especially about adhesive bonding (the principal joining process used with lightweight materials and dissimilar or multi-materials).

Author Response

Thank you very much for your time and helpful comments to improve the work. Please see the attachment.

Reviewer 2 Report

Dear Authors,

Your review of lightweight materials is quite comprehensive.

There are some minor issues that need to be addressed:

Line 70 – Paragraph about automotive industry emissions. You are stating that light duty vehicles are source of sulphur oxides which is very questionable. Also, your reference No7 in that paragraph guides to ICCT aviation website which is not related to road vehicles. Please check!

Line 228 – Boolean equation there is no explanation of abbreviation TS

Line 692 – Figure 8. a) Vertical axis title is missing. Figure 8b. is cropped on the right side.

Author Response

Thank you very much for your interest, time and helpful comments to improve the work. Please see the attachment.

Round 2

Reviewer 1 Report

It is clear now in the manuscript (form the introduction lines 171-180) the reason that the adhesive bonding was not included in the present study.

Please remove lines 948-958 (it is already clear from lines 171-180).

Author Response

Thank you very much for your review and contribution to the improvement of this work. Lines 948-958 have been removed.